# Mental Imagery in Fencing: Improving Point Control and Lunge Distance Through Visualization

**DOI:** 10.3390/brainsci15121338

**Published:** 2025-12-16

**Authors:** Troy Tianxing Song, Adam Liu, Kun Liu

**Affiliations:** 1Brain Peace Science Foundation, New Haven, CT 06511, USA; 2Department of Psychiatry, Yale University School of Medicine, New Haven, CT 06510, USA

**Keywords:** visualization, mental imagery, fencing, point control, lunge distance

## Abstract

**Background**: Visualization (motor imagery) is used in sports to enhance performance. Fencing relies on point control and lunge distance, yet little is known about how visualization affects these skills across experience levels. **Objective**: To examine the effects of brief visualization on point control and lunge distance in fencers of different experience levels. **Methods**: Nineteen fencers (age 10–56) completed pre- and post-tests of point control (10 hits) and lunge distance (maximum reach). Between tests, the experimental group performed a 1 min guided visualization, while the control group (*n* = 20) repeated the tests without visualization. **Results**: Visualization significantly improved point control (+1.3 hits, 25.5%; *p* = 0.002). Lunge distance increased (+15.6 cm, 11.1%; *p* = 0.001). Less experienced fencers improved more in point control (39.0% vs. 14.8%), while experienced fencers improved more in lunge distance (12.8% vs. 7.2%). Control participants showed no meaningful gains, and between-group comparisons confirmed significant advantages for visualization in both skills (*p* < 0.01). **Conclusion**: Even a short visualization exercise improved fencing performance, with novices benefiting most in accuracy and experienced fencers in explosive reach. Visualization offers a low-cost, adaptable supplement to fencing training.

## 1. Introduction

Visualization, or mental imagery, is a well-established technique in sports psychology that involves rehearsing movements mentally without physical execution. Research has shown that imagery can enhance accuracy, strength, reaction time, confidence, and overall performance by engaging neural pathways similar to those activated during actual movement [1,2]. Importantly, these benefits extend beyond elite athletes; novices also gain from mental imagery, which can accelerate skill acquisition and support motor learning [3,4]. Bibliometric analyses confirm the growing importance of psychological skills training, including visualization, within sports and exercise medicine [5]. Moreover, visualization is increasingly recognized as a core element of psychological skills training across athletic disciplines [6,7,8].

Despite broad application across sports, fencing remains underexplored in this context. Fencing demands a combination of precision and explosiveness, requiring fine control of the weapon tip and rapid execution of dynamic movements, such as the lunge. Both point control and lunge distance are central to successful performance, yet few studies have directly examined whether visualization can improve these fundamental skills in fencers. This is a notable gap, as fencing provides a unique model to study visualization’s effects on both fine motor accuracy and gross motor coordination [9].

The present study addresses this gap by testing the impact of a brief visualization exercise on point control and lunge distance in fencers. Additionally, it evaluates whether athletes with varying levels of experience respond differently to imagery. By combining controlled testing with subgroup comparisons, this study aims to clarify the role of visualization in fencing performance and provide practical insights for tailoring mental training across stages of athlete development.

## 2. Materials and Methods

The experiment consisted of two testing phases—pre-test and post-test—separated by a visualization exercise. Both phases included evaluation of point control and lunge distance. For a more detailed description of point control and lunge distance test set up and visualization procedures, refer to Appendix A.

### 2.1. Participants

Nineteen fencers aged 10 to 56 years participated in the experimental group. Participants were recruited from local fencing clubs and represented a diverse range of skill levels and years of training experience. All received a standardized explanation of the study procedures before testing [10,11]. A separate control group (*n* = 20) completed the same pre- and post-tests under identical conditions but did not perform the visualization exercise. This distinguished natural improvements during a retest versus visualization-specific effects.

### 2.2. Grouping

Participants were divided into two categories for subgroup analyses based on training history: less experienced fencers with <4 years of formal fencing training and more experienced fencers with ≥4 years of formal fencing training. This classification allowed investigation of whether visualization effects differed by training experience.

### 2.3. Pre-Test Procedures

Pre-Test procedures include point control test, lunge distance test and visualization exercise.

### 2.4. Point Control Test

Before the test, fencers were given one minute of practice to strike a target mounted on a wall. Four predetermined target locations were selected, each progressively increasing in difficulty [12,13]. During testing, participants attempted 10 hits on their designated target. The number of successful hits was recorded, and video recordings were used to verify accuracy. To avoid ceiling or floor effects, target difficulty was adjusted for subsequent attempts if necessary [10,14].

### 2.5. Lunge Distance Test

Participants began in the standard en garde position and executed a forward lunge as far as possible while maintaining balance and correct fencing form. The distance from the initial front foot position to the landing position was measured in centimeters. Each fencer performed three lunges, and the maximum distance was recorded for analysis [15].

### 2.6. Visualization Exercise

Following the pre-tests, participants in the experimental group engaged in a guided visualization session lasting approximately one minute in a quiet environment. They were instructed to close their eyes and mentally rehearse both tasks: point control and lunge imagery. In the point control imagery, fencers were asked to imagine extending the arm, aligning the blade, and striking the center of the target with precision. In the lunge imagery, fencers were asked to imagine pushing explosively off the back leg, extending the front leg forward, and maintaining balance through the landing position.

Participants were encouraged to use a first-person perspective, as this vantage point has been shown to provide behavioral advantages for motor learning [16]. Both kinesthetic (bodily sensations, muscle activation) and visual (seeing oneself perform successfully) imagery were emphasized. The exercise was standardized and led by the same instructor to ensure consistency.

### 2.7. Post-Test Procedures

Immediately after the visualization exercise, participants in the experimental group repeated the point control test and lunge distance test under identical conditions. Control group participants also repeated the tests, but without any visualization between sessions. This design allowed for direct within-group comparisons (pre vs. post) and between-group comparisons (experimental vs. control) [17,18].

### 2.8. Statistical Analysis

Data were analyzed using descriptive and inferential statistics. Pre- and post-test results for point control and lunge distance were compared within each group using appropriate statistical tests (see Appendix B for details). Differences between the experimental and control groups were evaluated using independent-sample *t*-tests. Subgroup analyses compared less experienced (<4 years) and more experienced (≥4 years) fencers using the same approach. Results are reported as means ± standard deviations, with percentage change calculated for each measure. Statistical significance was set at *p* < 0.05. All analyses were performed using Microsoft Excel 365 and verified with SPSS v29 (IBM Corp., Armonk, NY, USA).

## 3. Results

### 3.1. Demographics

Nineteen fencers from the LEO Fencing Club participated in the study, ranging from age 10 to 56 years. Years of fencing varied from none to 22 years of competitive practice. Of the participants, 78.9% were men and 21.1% were women. According to USA Fencing Age Classification [19], the distribution was: Y10 (5.3%), Y12 (10.5%), Y14 (15.8%), Cadet (26.3%), Junior (10.5%), Senior (15.8%), and Vet-50 (15.8%) (Table 1).

### 3.2. Overall Effects of Visualization

Across all participants, visualization training improved performance in both tasks.

Point Control: The average number of successful hits increased from 5.3 (pre-test) to 6.6 (post-test), representing an improvement of 1.3 hits (25.5%). This effect was statistically significant (*p* = 0.002) (Figure 1; Table 2; Appendix B). Prior research has emphasized the importance of attentional focus and tip accuracy in fencing performance [10,13].

Lunge Distance: The mean distance increased from 133.7 cm to 147.9 cm, corresponding to an improvement of 14.2 cm (11.1%). Strong significant improvement after visualization was observed (*p* = 0.001; Figure 2; Table 2; Appendix B). Previous biomechanical studies suggest that improvements in lunge distance reflect coordination and lower-limb power [15].

### 3.3. Effects of Training Experience

When fencers were divided by training history, additional patterns emerged.

Point Control: Fencers with <4 years of experience showed greater improvement, increasing from 5.1 to 7.1 hits (39.0%) (Table 3, Figure 3a). Those with ≥4 years of experience improved from 5.5 to 6.2 hits (14.8%) (Table 4, Figure 4a). These differences are consistent with evidence that novices benefit most from visualization in skill acquisition [3,4].Lunge Distance: Fencers with <4 years of experience improved by 8.9 cm (7.2%) (Table 5, Figure 3b). Fencers with ≥4 years of experience, improved by 18.1 cm (12.8%) (Table 6, Figure 4b). This aligns with previous findings that experienced fencers use imagery to refine explosive actions, such as lunges [20].

**Table 3 brainsci-15-01338-t003:** Point Control Data for Fencers with Less than Four Years of Training.

Fencer ID	Pre	Post	Difference	Percent Improvement
Fencer H	4	5	1	25.00%
Fencer M	4	5	1	25.00%
Fencer L	6	5	−1	−16.67%
Fencer D	5	7	2	40.00%
Fencer O	5	7	2	40.00%
Fencer B	7	9	2	28.57%
Fencer K	5	9	4	80.00%
Fencer A	5	10	5	100.00%
Average	5.1	7.1	2.0	39.0%

**Table 4 brainsci-15-01338-t004:** Point Control Data for Fencers with More than Four Years of Training.

Fencer ID	Pre	Post	Difference	Percent Improvement
Fencer P	4	4	0	0.00%
Fencer F	4	4	0	0.00%
Fencer N	6	6	0	0.00%
Fencer S	5	6	1	20.00%
Fencer C	5	6	1	20.00%
Fencer E	6	6	0	0.00%
Fencer G	4	7	3	75.00%
Fencer R	6	7	1	16.67%
Fencer I	6	7	1	16.67%
Fencer J	7	7	0	0.00%
Fencer Q	7	8	1	14.29%
Average	5.5	6.2	0.7	14.8%

**Table 5 brainsci-15-01338-t005:** Lunge Distance Data for Fencers with Less than Four Years of Training.

Fencer ID	Pre	Post	Difference	Percent Improvement
Fencer A	98	109	11	11.22%
Fencer H	95	111	16	16.84%
Fencer K	118	124	6	5.08%
Fencer M	126	128	2	1.59%
Fencer B	122	134	12	9.84%
Fencer L	127	145	18	14.17%
Fencer D	146	153	7	4.79%
Fencer O	156	155	−1	−0.64%
Average	123.5	132.4	8.9	7.2%

**Table 6 brainsci-15-01338-t006:** Lunge Distance Data for Fencers with More than Four Years of Training.

Fencer ID	Pre	Post	Difference	Percent Improvement
Fencer N	112	126	14	12.50%
Fencer I	123	134	11	8.94%
Fencer S	127	137	10	7.87%
Fencer C	123	143	20	16.26%
Fencer G	132	149	17	12.88%
Fencer P	146	150	4	2.74%
Fencer R	144	156	12	8.33%
Fencer Q	163	173	10	6.13%
Fencer F	177	186	9	5.08%
Fencer E	175	194	19	10.86%
Fencer J	131	204	73	55.73%
Average	141.2	159.3	18.1	12.8%

### 3.4. Control Group

The control group performed the same pre- and post-tests without visualization training. Results showed minimal or negative changes:Point Control: Average scores declined slightly from 6.1 to 5.75 hits (−0.35 hits; −6.6%) (Table 7).Lunge Distance: Distances increased only marginally, from 123.4 cm to 126.7 cm (+3.31 cm; 2.76%) (Table 7).

### 3.5. Between Group Comparisons

Comparisons between the visualization and control groups showed significant differences (Figure 5).

Point Control: The visualization group showed significantly greater improvement than the control group (Mann–Whitney U test, *p* = 0.0057). This confirms that visualization training, rather than test repetition, accounted for the gains [9,21].Lunge Distance: Improvements in the visualization group were also significantly greater than those in the control group (Welch *t*-test, *p* = 0.0072).

## 4. Discussion

This study investigated whether a brief visualization exercise could enhance two key fencing skills—point control and lunge distance—and whether these effects varied by level of training experience. The findings indicate that visualization enhances fencing performance, but the nature of these benefits depends on both the fencer’s level of expertise and the type of technical skill.

### 4.1. Visualization and Point Control

Visualization significantly improved point control, with fencers increasing their average number of successful strikes by 25.5%. This effect was most pronounced in less experienced athletes (<4 years), who improved by 39.0% compared to only 14.8% in more experienced athletes. These results support the idea that novices, who are still forming stable motor representations unique to sport, gain more from mental rehearsal of fine motor skills. Imagery strengthens neural pathways associated with precision movements, enabling athletes to consolidate technical accuracy even without physical practice [3,4,22].

### 4.2. Visualization and Lunge Distance

Unlike point control, improvements in lunge distance were greater among more experienced fencers. While novices showed modest gains (7.2%), experienced fencers improved their reach by an average of 12.8%. These findings align with prior research suggesting that athletes with well-established motor schemas can utilize imagery to refine explosive movements by enhancing proprioceptive awareness, muscular coordination, and timing [5,20].

### 4.3. Comparison with Control Group

The inclusion of a control group confirmed that the observed benefits were not due to natural improvements from repeated testing. Control participants showed no meaningful gains in point control and only minimal improvement in lunge distance, in contrast to the significant improvements in the visualization group. The mean number of successful baseline hits prior to visualization was 5.3 ± 1.03 for the test group and 6.1 ± 2.0 for the control group. Likewise, the mean baseline lunge distance was 133.7 ± 22.2 cm in the test group and 123.4 ± 21.6 cm in the control group. Based on these mean values and associated standard deviations, no statistically significant differences were observed between groups at baseline, indicating that the test and control groups were comparable before the intervention. This confirms the conclusion that visualization was the key factor in driving performance gains. Similar patterns have been observed in other sports, where even brief imagery interventions led to measurable improvements in skill performance [9,21].

### 4.4. Practical Applications

These findings highlight the value of tailoring visualization exercises to the athlete’s developmental stage. For novices, emphasizing imagery of precise blade control may accelerate the acquisition of technical skills. For experienced athletes, focusing on explosive and dynamic actions such as the lunge may refine already well-developed motor patterns. Importantly, visualization requires no equipment and is time-efficient, making it an accessible tool for daily training sessions.

### 4.5. Limitations and Future Directions

Several limitations should be acknowledged. The sample size was modest despite the fact that both point control and lunge distance improvements reached statistical significance. Visualization sessions were standardized and brief, which may not fully account for individual differences in imagery ability. Previous research has shown that interference during imagery can influence outcomes, highlighting the importance of considering imagery quality and cognitive load [23]. Furthermore, as with all small-sample experimental designs, results must be interpreted cautiously in light of potential biases and analytic challenges [24]. Future research should explore longer-term interventions, personalized imagery scripts, and interactions with psychological variables such as confidence and attentional control. Neuroimaging studies could further clarify the neural mechanisms of visualization in fencing, building on past research that demonstrates sport-specific brain adaptations [11,14,25].

## 5. Conclusions

Overall, this study provides evidence that even short visualization exercises can improve fencing performance [26,27]. The specific benefits vary with training experience—point control gains being most prominent in novices and lunge distance gains in experienced athletes. By integrating targeted visualization into practice routines, coaches may be able to accelerate skill acquisition, optimize performance, and complement traditional physical training methods.

## Figures and Tables

**Figure 1 brainsci-15-01338-f001:**
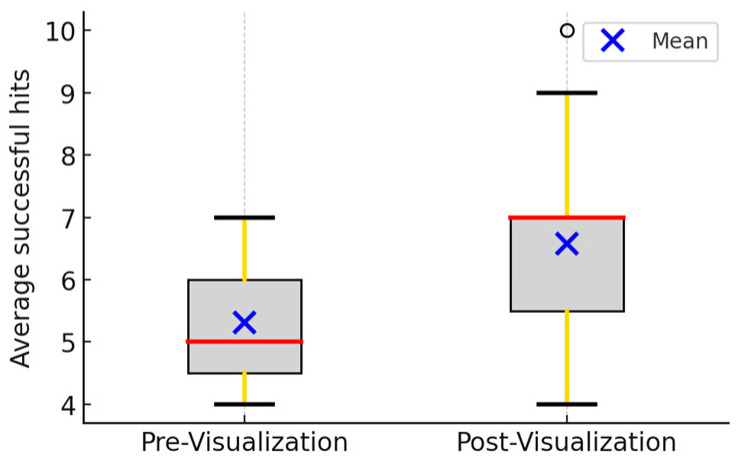
Average successful hits among 19 fencers. The left bar represents the results from pre-visualization, while the right bar represents the results from post-visualization. Red lines represent the median values. Wilcoxon Signed-Rank test *t*-test (*p*-value: 0.002). The black lines represent maximum and minimum non-outlier values in the data set. Outliers are represented by black circles.

**Figure 2 brainsci-15-01338-f002:**
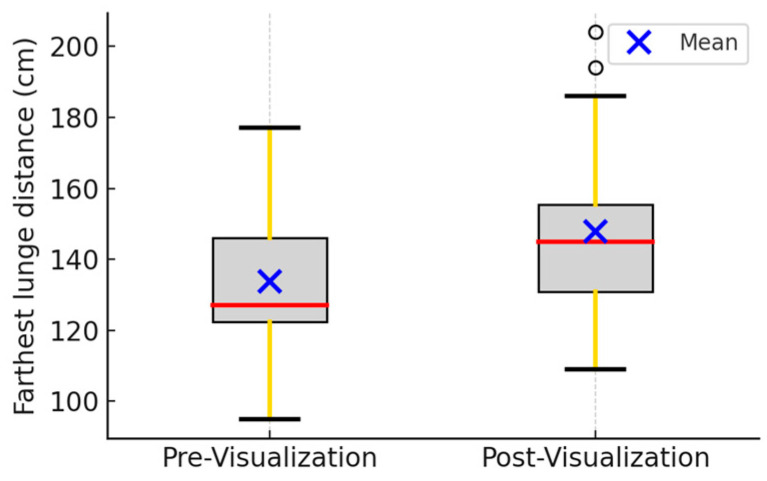
Average distance among 19 fencers. The left bar represents the results from pre-visualization, while the right bar represents the results from post-visualization. Error bars are the standard deviation. Red lines represent the median values. The black lines represent maximum and minimum non-outlier values in the data set. Error bars are the standard deviation. Paired *t*-test (*p*-value: 0.001). Outliers are represented by black circles.

**Figure 3 brainsci-15-01338-f003:**
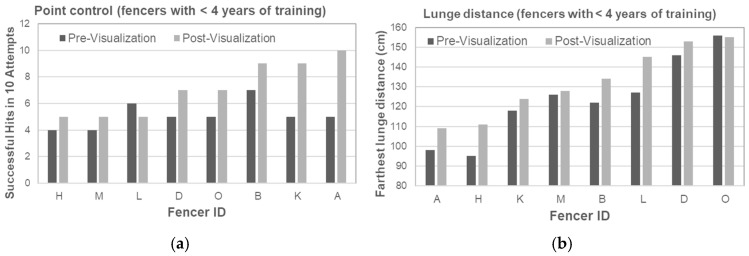
(**a**) Point Control Analysis in Fencers with Less than Four Years of Training. (**b**) Lunge Distance Analysis in Fencers with Less than Four Years of Training.

**Figure 4 brainsci-15-01338-f004:**
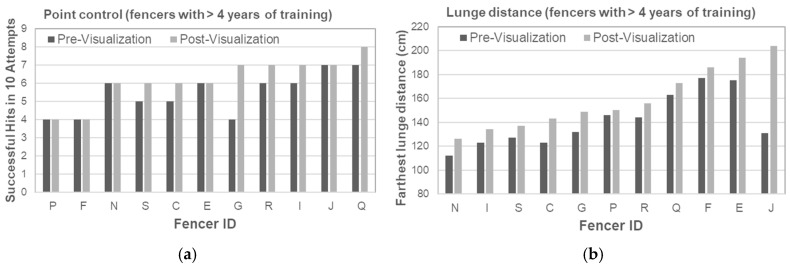
(**a**) Point Control Analysis in Fencers with More than Four Years of Training. (**b**) Lunge Distance Analysis in Fencers with More than Four Years of Training.

**Figure 5 brainsci-15-01338-f005:**
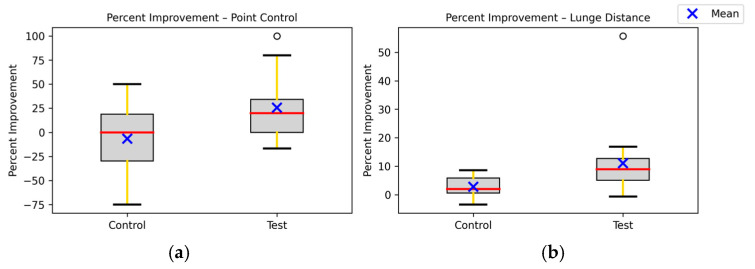
The Test group shows a much higher average improvement in both metrics: point control and lunge distance. The red lines in the box plots represent the median; The black lines represent maximum and minimum non-outlier values in the data set; Blue Xs represent the mean, and gray boxes represent the interquartile range (IQR). (**a**) The percentage improvement between the Control Group and Test Group in the Point Control test. The Control group has wide variability and several negative performances in Point Control. Mann–Whitney U test (*p*-value: 0.0057). (**b**) Average percentage improvement between the Control Group and Test Group in the Lunge Distance test. Welch *t*-test (*p*-value: 0.0072). The black circles represent the statistical outliers in the graph.

**Table 1 brainsci-15-01338-t001:** Distribution of age, years of fencing experiences, gender and USA fencing age classification among control and test groups. The fencing age classification is based on the rules determined by the USA Fencing [19].

Fencer ID	Age	USA Fencing Age Category	Gender	Years of Fencing Experience	Group
Control A	12	Y12	Male	4	Control
Control B	17	Junior	Male	6	Control
Control C	12	Y12	Female	4	Control
Control D	11	Y12	Female	3	Control
Control E	16	Cadet	Male	8	Control
Control F	16	Cadet	Male	8	Control
Control G	17	Junior	Female	8	Control
Control H	16	Cadet	Male	8	Control
Control I	12	Y12	Male	6	Control
Control J	15	Cadet	Male	10	Control
Control K	15	Cadet	Male	7	Control
Control L	16	Cadet	Male	5	Control
Control M	50	Vet-50	Male	0	Control
Control N	47	Vet-40	Female	0	Control
Control O	12	Y12	Male	4	Control
Control P	16	Cadet	Male	7	Control
Control Q	14	Y14	Male	4	Control
Control R	51	Vet-50	Male	2.5	Control
Control S	15	Cadet	Male	4	Control
Control T	13	Y14	Female	4	Control
AVE	20	-	-	5	-
STDEV	13	-	-	3	-
Test A	54	Vet-50	Men	0.5	Test
Test B	15	Cadet	Women	2	Test
Test C	17	Junior	Men	11	Test
Test D	15	Cadet	Men	4	Test
Test E	19	Junior	Men	12	Test
Test F	32	Senior	Men	22	Test
Test G	12	Y12	Women	5	Test
Test H	57	Vet-50	Men	0	Test
Test I	27	Senior	Men	15	Test
Test J	31	Senior	Men	20	Test
Test K	14	Y14	Men	4	Test
Test L	12	Y12	Men	4	Test
Test M	13	Y14	Women	2.5	Test
Test N	10	Y10	Men	5	Test
Test O	14	Y14	Men	4	Test
Test P	16	Cadet	Men	7	Test
Test Q	15	Cadet	Men	10	Test
Test R	56	Vet-50	Men	5	Test
Test S	14	Cadet	Women	6	Test
AVE	23	-	-	7	-
STDEV	16	-	-	6	-

**Table 2 brainsci-15-01338-t002:** Results of point control and lunge distance tests before and after visualization. The calculated averages are shown at the bottom. The difference is the post-data minus pre-data. The definition of the “%” in the table is the percent improvement between “pre” and “post”.

Fencer ID	Point Control				Lunge Distance (cm)			
	Pre	Post	Difference	%	Pre	Post	Difference	%
A	5	10	5	100.00%	98	109	11	11.22%
B	7	9	2	28.57%	122	134	12	9.84%
C	5	6	1	20.00%	123	143	20	16.26%
D	5	7	2	40.00%	146	153	7	4.79%
E	6	6	0	0.00%	175	194	19	10.86%
F	4	4	0	0.00%	177	186	9	5.08%
G	4	7	3	75.00%	132	149	17	12.88%
H	4	5	1	25.00%	95	111	16	16.84%
I	6	7	1	16.67%	123	134	11	8.94%
J	7	7	0	0.00%	131	204	73	55.73%
K	5	9	4	80.00%	118	124	6	5.08%
L	6	5	−1	−16.67%	127	145	18	14.17%
M	4	5	1	25.00%	126	128	2	1.59%
N	6	6	0	0.00%	112	126	14	12.50%
O	5	7	2	40.00%	156	155	−1	−0.64%
P	4	4	0	0.00%	146	150	4	2.74%
Q	7	8	1	14.29%	163	173	10	6.13%
R	6	7	1	16.67%	144	156	12	8.33%
S	5	6	1	20.00%	127	137	10	7.87%
Average	5.3	6.6	1.3	25.50%	133.7	147.9	14.2	11.06%

**Table 7 brainsci-15-01338-t007:** Results of point control and lunge distance tests in the control group. The calculated averages are shown at the bottom. The difference is the post data minus pre data. The definition of the “%” in the table is the percent improvement between “pre” and “post”.

Fencer ID	Point Control				Lunge Distance (cm)			
	First	Second	Difference	%	First	Second	Difference	%
A	5	3	−2	−40.00%	129	132	3	2.33%
B	6	5	−1	−16.67%	156	165	9	5.77%
C	7	6	−1	−14.29%	122	124	2	1.64%
D	7	5	−2	−28.57%	128	131	3	2.34%
E	4	1	−3	−75.00%	146	157	11	7.53%
F	3	3	0	0.00%	130	131	1	0.77%
G	7	4	−3	−42.86%	110	109	−1	−0.91%
H	4	4	0	0.00%	140	148	8	5.71%
I	4	2	−2	−50.00%	102	101	−1	−0.98%
J	3	4	1	33.33%	119	120	1	0.84%
K	6	7	1	16.67%	139	141	2	1.44%
L	6	9	3	50.00%	141	143	2	1.42%
M	6	4	−2	−33.33%	101	108	7	6.93%
N	4	5	1	25.00%	81	86	5	6.17%
O	7	9	2	28.57%	120	119	−1	−0.83%
P	10	9	−1	−10.00%	155	160	5	3.28%
Q	8	8	0	0.00%	114	122	8	6.67%
R	8	8	0	0.00%	89	97	8	8.57%
S	8	10	2	25.00%	147	142	−5	−3.45%
T	9	9	0	0.00%	99	99	0	0.00%
Average	6.1	5.75	−0.35	−6.61%	123.4	126.7	3.31	2.76%

## Data Availability

The datasets generated and analyzed during the current study are available from the corresponding author on reasonable request. Due to privacy considerations, individual fencer identifiers have been anonymized.

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
