# Peer review of "Mental Imagery in Fencing: Improving Point Control and Lunge Distance Through Visualization"

_brainsci, 2025, doi:10.3390/brainsci15121338_

Round 1

Reviewer 1 Report

Comments and Suggestions for Authors

I consider the manuscript well-constructed and addressing an important topic in sports psychology, with valuable data on fencing, which is relatively under-researched in the context of mental imagery. The study is interesting due to its division into experienced and less experienced fencers and the short duration of the intervention.

This study fills a gap in the literature on the use of visualization in fencing by focusing on two key skills: fine motor accuracy and gross motor coordination/explosiveness. The findings regarding differential benefits for novice and experienced fencers are important and consistent with general theories of skill acquisition.

The following comments are constructive and should help refine the work before publication.

  1. Visualization covered both point control and lunge, and fencers were encouraged to use a first-person perspective and a combination of kinesthetic and visual imagery. This description is adequate, but please consider including the full script or a more detailed summary in the Appendix for full replicability.
  2. The authors report that the visualization session lasted approximately five minutes (CHAPTER 2.6. Visualization Exercise). The abstract states that the experimental group completed a 1-minute visualization. This discrepancy (1 minute vs. 5 minutes) in the length of the intervention needs to be clarified/corrected.
  3. The language is clear and requires no significant corrections. There is a minor typo in the title of Figure 3 on page 4, where "Lunge" is written as "Lfarthest young Distance (cm)". This should be corrected.
  4. The authors are strongly advised to incorporate at least three-five additional, targeted references to enhance the manuscript's scholarly depth, particularly concerning the efficacy of short interventions, and the theoretical underpinnings of experience-dependent skill acquisition.

See: Schuster C, Hilfiker R, Amft O, et al. Best practice for motor imagery: a systematic literature review on motor imagery training elements in five different disciplines. BMC Med. 2011;9:75. Published 2011 Jun 17. doi:10.1186/1741-7015-9-75

Author Response

Comment 1: Visualization covered both point control and lunge, and fencers were encouraged to use a first-person perspective and a combination of kinesthetic and visual imagery. This description is adequate, but please consider including the full script or a more detailed summary in the Appendix for full replicability.

Response 1: Thank you for your suggestion. We agree that providing more detail enhances the replicability of our study. A significantly expanded and detailed description of the visualization procedure has been added to Appendix A.

Comment 2: The authors report that the visualization session lasted approximately five minutes (CHAPTER 2.6. Visualization Exercise). The abstract states that the experimental group completed a 1-minute visualization. This discrepancy (1 minute vs. 5 minutes) in the length of the intervention needs to be clarified/corrected.

Response 2: We apologize for the discrepancy. The correct duration of the visualization session is one minute. This has now been corrected throughout the manuscript.

Comment 3: The language is clear and requires no significant corrections. There is a minor typo in the title of Figure 3 on page 4, where "Lunge" is written as "Lfarthest young Distance (cm)". This should be corrected.

Response 3: Thank you for spotting this typo. The original box plot has been replaced with a revised version to match the formatting of other figures. The y-axis label has been corrected to “farthest lunge distance (cm)”.

Comment 4: The authors are strongly advised to incorporate at least three-five additional, targeted references to enhance the manuscript's scholarly depth, particularly concerning the efficacy of short interventions, and the theoretical underpinnings of experience-dependent skill acquisition.

Response 4: Thank you for this valuable recommendation. We have incorporated three additional references that focus on the efficacy of short interventions and the theoretical foundations of experience-dependent skill acquisition. For example, we now cite Schuster C, Hilfiker R, Amft O, et al. (2011), Best practice for motor imagery: a systematic literature review on motor imagery training elements in five different disciplines, BMC Med, 9:75. doi:10.1186/1741-7015-9-75, among others.

Reviewer 2 Report

Comments and Suggestions for Authors

Dear authors,

Mental Imagery in Fencing: Improving Point Control and Lunge Distance Through Visualization article was well written and a simple straight forward approach. Having the fencers take metal reps with specific cues is a good visualization approach. The intro was concise and to the point, the methods where easy to understand and the results were presented. The results section needs the most work as Figure one would be better suits as a table instead of a histogram and pie chart. Figures 2 and 3 need to match the same formatting as the rest of the paper. Table 1. Data point for L in the percent column needs a value and needs to be all on the same page. Figures 4 and 5need to be cleaned up as you cannot read what the green and grey means. Overall the results section needs to be more visually appealing and easier to understand. Discussion is adequate and Conclusion is based on the results.

Author Response

Comment 1: The results section needs the most work as Figure one would be better suits as a table instead of a histogram and pie chart. 

Response 1: Thank you for this helpful suggestion. In response, we have replaced the histograms and pie charts with a table that presents the data more clearly. This updated table now includes data from both the experimental and control groups.

Comment 2: Figures 2 and 3 need to match the same formatting as the rest of the paper. 

Response 2: Thank you for pointing this out. We have revised Figures 2 and 3 to ensure they are consistent in style and formatting with the other figures throughout the manuscript.

Comment 3: Table 1. Data point for L in the percent column needs a value and needs to be all on the same page. 

Response 3: We appreciate your attention to detail. The missing data for Fencer L was due to formatting limitations. This has now been corrected—Fencer L had a -16.67% change. Additionally, we have adjusted the layout so that Table 1 now appears fully on a single page.

Comment 4: Figures 4 and 5need to be cleaned up as you cannot read what the green and grey means.

Response 4: Thank you for highlighting this issue. We have revised Figures 4 and 5 by adjusting the bar colors to match the scheme used in other figures, increasing the legend font size, and adding clear chart titles to improve readability.

Reviewer 3 Report

Comments and Suggestions for Authors

Modern sport is a field of activity that captivates not only the participants themselves but also the spectators. Professional sport requires maximum concentration on the sports result, which is determined by many components. Technical sports are especially difficult in this area. In this case, the research suggested by the authors is aimed at finding methods of sports training that can increase the effectiveness of individual techniques in fencing. Ideomotor training is an important component of educational and training process and is most widely implemented in technical sports, however, there are still many unexplored issues of its application.

The authors have constructed a structured, consistent presentation of information that allows for the analysis of obtained results and the verification of formulated conclusions.

Regarding possible verification of the hypothesis put forward, there are a number of issues that require additional justification in the article.

The authors describe in detail the methodology for conducting the study, which allows it to be reproduced if necessary. The article is well illustrated, which makes it easy to analyze the presented data.

However, a number of questions arise while reading the article that require additional explanation:

In Figure 1, please, add information regarding which group the distribution is presented, and why information on the control group is missing.

How did the participants of the control and experimental groups compare in terms of qualification level and gender?

Were there differences in the results of test exercises among participants in the experimental and control groups before the experiment?

In section "2.8. Statistical Analysis" you should provide information on checking the samples for a normal distribution, justifying why t-tests were used to determine statistically significant differences.

In section "2.5. Lunge Distance Test" more information should be provided regarding description of this test, who proposed this test or whether it is an author's, the method of evaluating the results, describe whether a motor action was performed on the dominant limb. The question arises of appropriateness of using absolute values ​​when analyzing the results of this test due to the fact that the study participants belong to radically different age groups from 10 years to 56 years and have different lengths of the lower limb.

In the discussion, more attention should be paid to comparing the obtained results and those of other authors.

Authors need to pay more attention to research by other authors over the last 5 years.

Author Response

Comment 1: In Figure 1, please, add information regarding which group the distribution is presented, and why information on the control group is missing.

Response 1: Thank you for your observation. The original figures have been converted into a table based on a request from another reviewer. In this revised table, data from both the experimental and control groups are now included. The group to which each distribution belongs is clearly indicated, and the previously missing control group data has been added for completeness.

Comment 2: How did the participants of the control and experimental groups compare in terms of qualification level and gender?

Response 2: Thank you for your comment. We have updated the demographic table to include detailed information on the qualification levels and gender distribution for both the control and experimental groups. This addition provides a clearer basis for comparing the two groups in terms of their background characteristics.

Comment 3: Were there differences in the results of test exercises among participants in the experimental and control groups before the experiment?

Response 3:  Thank you for your question. There were no statistically significant differences between the experimental and control groups in either average hits or lunge distance during the pre-visualization tests. Specifically, the test group had an average of 5.3 hits (SD = 1.03) and an average lunge distance of 133.7 cm (SD = 22.2 cm), while the control group averaged 6.1 hits (SD = 2.0) and a lunge distance of 123.4 cm (SD = 21.6 cm). These pre-intervention values indicate comparable baseline performance between the two groups. A comparative summary of these results has been added to the Discussion section under Comparison with Control Group.

Comment 4: In section "2.8. Statistical Analysis" you should provide information on checking the samples for a normal distribution, justifying why t-tests were used to determine statistically significant differences.

Response 4: Thank you for your observation. We have now evaluated the normality of the sample distributions using Q-Q plots for each dataset that was previously associated with a p-value. These Q-Q plots have been added to Appendix B for reference. Based on the visual inspection of these plots, we determined whether the assumption of normality was met. Accordingly, we revised the statistical tests where necessary: parametric tests (e.g., t-tests) were used when the data appeared normally distributed, while non-parametric alternatives were applied where the normality assumption was not supported. All graphs now display the appropriate statistical test alongside the updated p-values.

Comment 5: In section "2.5. Lunge Distance Test" more information should be provided regarding description of this test, who proposed this test or whether it is an author's, the method of evaluating the results, describe whether a motor action was performed on the dominant limb. The question arises of appropriateness of using absolute values ​​when analyzing the results of this test due to the fact that the study participants belong to radically different age groups from 10 years to 56 years and have different lengths of the lower limb.

Response 5: Thank you for reviewer's valuable comment. We have now added a comprehensive description of the Lunge Distance Test to Appendix A, including details on its origin, procedures, method of measurement, and whether the dominant limb was used. This should clarify the test's design and execution. We also fully acknowledge the concern regarding the appropriateness of using absolute values when participants vary significantly in age and limb length. Indeed, these factors could influence performance outcomes. While our current analysis focused on absolute lunge distances, we observed age-related differences in the response to motor imagery interventions, suggesting that lower limb length and developmental stage may interact with the effectiveness of visualization. However, our current study was not specifically designed to evaluate limb length or age as moderating variables in the visualization effect. We agree that a normalized measure (e.g., relative to limb length or height) might improve interpretability and comparability across age groups. This is an important consideration, and we plan to incorporate these adjustments in a future study aimed at more precisely examining how anthropometric and developmental factors influence motor imagery efficacy.

Comment 6: In the discussion, more attention should be paid to comparing the obtained results and those of other authors. Authors need to pay more attention to research by other authors over the last 5 years.

Response 6: Thank you for this important suggestion. In response, we have revised the Discussion section to include comparisons between our findings and those of other researchers. Specifically, we have added three additional references, two of which are from the last five years, to strengthen the relevance and contextual grounding of our results within the current literature. These updates help to highlight both consistencies and distinctions between our findings and recent studies on motor imagery and motor performance.

Round 2

Reviewer 3 Report

Comments and Suggestions for Authors

the article may be recommended for publication